# Multimodal vaccination targeting the receptor binding domains of *Clostridioides difficile* toxins A and B with an attenuated *Salmonella* Typhimurium vector (YS1646) protects mice from lethal challenge

Kaitlin Winter,[1,2] Sébastien Houle,[3] Charles M. Dozois,[3] Brian J. Ward[1,2]

**ABSTRACT**  Developing a vaccine against *Clostridioides difficile* is a key strategy to protect the elderly. Two candidate vaccines using a traditional approach of intramuscular (IM) delivery of recombinant antigens targeting *C. difficile* toxins A (TcdA) and B (TcdB) failed to meet their primary endpoints in large phase 3 trials. To elicit a mucosal response against *C. difficile*, we repurposed an attenuated strain of *Salmonella* Typhimurium (YS1646) to deliver the receptor binding domains (rbd) of TcdA and TcdB to the gut-associated lymphoid tissues, to elicit a mucosal response against *C. difficile*. In this study, YS1646 candidates with either rbdA or rbdB expression cassettes integrated into the bacterial chromosome at the *att*Tn7 site were generated and used in a short-course multimodal vaccination strategy that combined oral delivery of the YS1646 candidate(s) on days 0, 2, and 4 and IM delivery of recombinant antigen(s) on day 0. Five weeks after vaccination, mice had high serum IgG titers and increased intestinal antigen-specific IgA titers. Multimodal vaccination increased the IgG avidity compared to the IM-only control. In the mesenteric lymph nodes, we observed increased IL-5 secretion and increased IgA+ plasma cells. Oral vaccination skewed the IgG response toward IgG2c dominance (vs IgG1 dominance in the IM-only group). Both oral alone and multimodal vaccination against TcdA protected mice from lethal *C. difficile* challenge (100% survival vs 30% in controls). Given the established safety profile of YS1646, we hope to move this vaccine candidate forward into a phase I clinical trial.

**IMPORTANCE**  *Clostridioides difficile* remains a major public health threat, and new approaches are needed to develop an effective vaccine. To date, the industry has focused on intramuscular vaccination targeting the *C. difficile* toxins. Multiple disappointing results in phase III trials have largely confirmed that this may not be the best strategy. As *C. difficile* is a pathogen that remains in the intestine, we believe that targeting mucosal immune responses in the gut will be a more successful strategy. We have repurposed a highly attenuated *Salmonella* Typhimurium (YS1646), originally pursued as a cancer therapeutic, as a vaccine vector. Using a multimodal vaccination strategy (both recombinant protein delivered intramuscularly and YS1646 expressing antigen delivered orally), we elicited both systemic and local immune responses. Oral vaccination alone completely protected mice from lethal challenge. Given the established safety profile of YS1646, we hope to move these vaccine candidates forward into a phase I clinical trial.

**KEYWORDS**  *Clostridioides difficile*, *Salmonella* Typhimurium, vaccination, chromosomal integration, mucosal immunity, humoral immunity

Address correspondence to Brian J. Ward, brian.ward@mcgill.ca.

K.W. and B.J.W. are inventors on a patent for YS1646 as a vaccine against *Clostridioides difficile* held by Aviex Technologies LLC. S.H. and C.M.D. have no conflict of interest to declare.

See the funding table on p. 16.

*C*lostridioides difficile infection (CDI) is one of the most important nosocomial infections. In the United States in 2017, CDI led to 223,900 hospitalized cases and 12,800 deaths, with an estimated cost of close to $6 billion (1, 2). In part due to better decontamination protocols and transmission control, hospital-acquired CDI rates have been dropping since 2009 (3, 4). However, two thirds of cases occur in patients over the age of 65, and the number of elderly in the United States is expected to double by 2050 (5–7). The highest risk for CDI is during administration of antibiotics and during the first month after cessation (6). Paradoxically in many respects, current treatment strategies for CDI are based primarily on prescribing additional antibiotics, and up to 35% of the patients who initially recover experience disease recurrence within the following 3–6 months (8). While fecal microbiome therapy is a highly publicized second-line treatment option, it is cumbersome, is not widely available, and has its own spectrum of complications (6). Prevention, in the form of transmission control, is a key strategy in reducing the CDI disease burden, but vaccination to either prevent disease or to reduce the rate of recurrence would be another powerful tool (7, 9, 10).

CDI is a toxin-mediated disease; most strains produce two main cytotoxins: toxin A (TcdA) and toxin B (TcdB) (11). Both toxins contribute to disease and irreversibly glycosylate Rho GTPases in intestinal epithelial cell cytosol leading to the disruption of the cytoskeleton and tight junctions, loss of stress fibers, and an overall loss of intestinal barrier function (12–17). The toxins are immunogenic, and anti-toxin antibodies have strong neutralization activity *in vitro* (18). Currently, the three vaccines that have reached the stage of phase 2/3 clinical trials have targeted both toxins using intramuscular (IM) vaccination with adjuvants to generate a systemic immune response (19–22). Two of these candidate vaccines have failed to meet their primary endpoints in phase III, and further development of one was formally abandoned (23, 24). These failures near the end of development pipeline suggest that novel strategies are needed in the design of vaccines for *C. difficile*. Since *C. difficile* is an extracellular and non-invasive pathogen of the gut, we reasoned that a vaccine capable of generating both systemic and mucosal immunity might be more successful in targeting the *C. difficile* toxins in the gut lumen.

*Salmonella enterica* has been investigated as a potential live-attenuated vaccine vector for decades (25). The success of *S. enterica* serovar Typhi (*S*. Typhi) Ty21a as a live-attenuated vaccine against typhoid in the 1980s clearly demonstrated the ability of attenuated *Salmonella* to induce a protective response (26). While wild-type *S*. Typhi causes a systemic infection, *Salmonella enterica* serovar Typhimurium (*S*. Typhimurium) is restricted to the gastrointestinal tract in humans (27). Like all *Salmonella* species, *S*. Typhimurium uses its type 3 secretion systems (T3SS) to maintain an intracellular life cycle in host intestinal epithelial cells and macrophages (28). Early effector proteins secreted by the *Salmonella* pathogenicity island I (SPI-I) T3SS promote entry into host cells (28). Once inside the cell and relatively protected inside a *Salmonella*-containing vacuole (SCV), proteins secreted by a second T3SS, SPI-II, are used to maintain the SCV (28). The SPI-I and SPI-II T3SSs can be co-opted in vaccine design to deliver heterologous proteins in the gut and adjacent immune tissues by attenuated strains of *S*. Typhimurium (29).

Our group is repurposing a strain of *S*. Typhimurium, YS1646, which was originally designed as a possible cancer therapeutic in the late 1980s. Compared to its parental strain (YS72), YS1646 has an *msbB⁻* mutation that leads to production of a modified lipopolysaccharide mutation with markedly reduced potential to cause septic shock (30–33). After extensive testing in multiple animal models, YS1646 was used in a large phase I clinical trial in subjects with advanced cancer (34). Although it failed as an anti-cancer "drug," this research demonstrated that YS1646 was safe even when injected at doses up to $10^8$ CFU intravenously (34). In early work using a plasmid-based antigen expression system, we developed two main vaccine candidates that expressed a portion of the receptor binding domains (rbd) of TcdA and TcdB (35). When delivered in a multimodal vaccination strategy (i.e., oral dosing combined with recombinant antigen

delivered intramuscularly), these vaccine candidates were able to protect mice from lethal challenge with *Clostridioides difficile* (35).

Since these YS1646 strains contain an antibiotic resistance gene on a mobile genetic element, in this work, we sought to produce suitable vaccine candidates by stably integrating the genes of our antigens into the bacterial genome at the *att*Tn*7* site on the bacterial chromosome. After screening multiple candidates with integrated *tcdA* or *tcdB* gene domains fused to sequences encoding a type 3 secretion signal and testing expression from different promoters, we chose two strains that had detectable antigen expression in Luria broth (LB) and tested them in a multimodal vaccination strategy in mice. We observed that IM vaccination elicited high IgG titers, with multimodal vaccination increasing the avidity of TcdB-specific antibodies. Oral vaccination alone and multimodal vaccination against TcdA completely protected mice from *C. difficile* challenge.

## RESULTS

### *S.* Typhimurium YS1646 expresses recombinant proteins from chromosomally integrated genes

To generate stably expressing recombinant antigens from genes integrated into the *Salmonella* Typhimurium YS1646 chromosome, six different sets of promoter, secretion signal, and antigen sequence combinations were designed and integrated into the *att*Tn*7* site (Table 1). Combinations were selected based on previous research using plasmid-based antigen expression (35). The promoters used were either constitutively active ($P_{frr}$) or induced when *Salmonella* is within host cells ($P_{pagC}$ and $P_{sspH2}$) (36–39). Two secretory signals were used that drive secretion at different times during bacterial invasion and intracellular residence. SspH2 is specific to the SPI-II T3SS, while SspH1 can be secreted by both SPI-I and SPI-II T3SSs (39). The same regions encoding the receptor binding domain of TcdA (rbdA) or TcdB (rbdB) that were used in our plasmid-based vaccines were used as recombinant antigens (35). All primers used in the study are listed in the supplemental materials. Each sequence for insertion was first placed in the pGp-Tn*7*-Cm plasmid, a mobilizable, pir-dependent suicide vector (Fig. 1A). Through bacterial conjugation, the pGp-Tn*7*-Cm-derived plasmids were introduced to YS1646 cells containing the Tn*7* transposase vector on a temperature-sensitive plasmid. Chromosomal integration occurred at the *att*Tn*7* site which is located adjacent to the constitutively active *glmS* gene (locus_tag: SL1344_3828; FQ312003) (40). After successful integration, FRT recombination was used to remove the chloramphenicol resistance cassette, generating YS1646 containing the desired recombinant sequences of interest stably integrated within the bacterial chromosome in the absence of any antibiotic-resistant gene (Fig. 1B). Successful integration was confirmed by PCR and whole genome sequencing (Files S2 to S4).

*In vitro* antigen expression in the vaccine candidates was evaluated by Western blotting. Two candidates (i.e., frr_SspH1_rbdA and pagC_SspH1_rbdB) expressed detectable levels of the target antigens following growth *in vitro* (Fig. 1C). The increased size of the *Salmonella*-expressed antigens is consistent with the secretory signals that are not present in the positive controls. By contrast, the four other strains generated did not

**TABLE 1** YS1646 strains used in this study

| Strain | Promoter | Secretory signal | Antigen |
|---|---|---|---|
| YS1646 | | | |
| frr_SspH1_rbdA (YS1646::rbdA) | $P_{frr}$ | SspH1 | TcdA$_{1820-2710}$ |
| pagC_SspH1_rbdA | $P_{pagC}$ | SspH1 | TcdA$_{1820-2710}$ |
| SspH2_SspH2_rbdA | $P_{sspH2}$ | SspH2 | TcdA$_{1820-2710}$ |
| frr_SspH1_rbdB | $P_{frr}$ | SspH1 | TcdB$_{1821-2366}$ |
| pagC_SspH1_rbdB (YS1646::rbdB) | $P_{pagC}$ | SspH1 | TcdB$_{1821-2366}$ |
| SspH2_SspH2_rbdB | $P_{sspH2}$ | SspH2 | TcdB$_{1821-2366}$ |

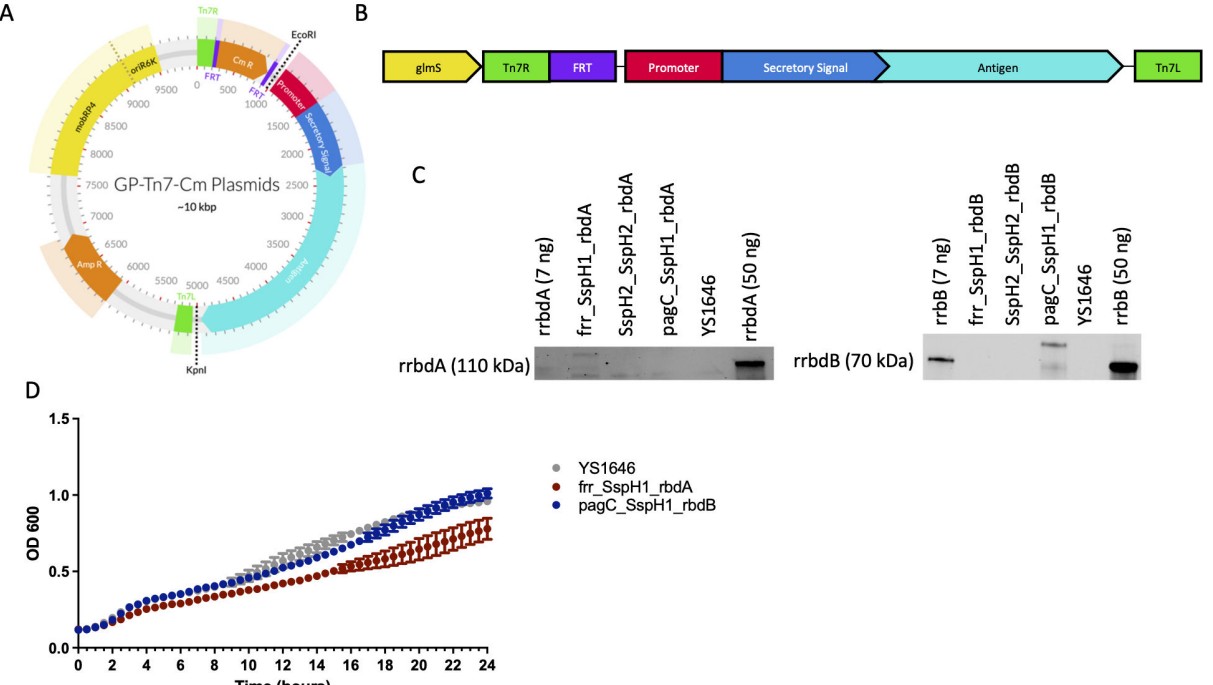

FIG 1 *S.* Typhimurium YS1646 expresses recombinant proteins from chromosomally integrated genes. (A) The promoter and regions encoding the secretory signal and antigen sequence were inserted into ampicillin- and chloramphenicol-resistant, mobilizable, and *pir*-dependent suicide vector pGp_Tn*7* plasmids using the EcoRI and KpnI restriction sites and uni seamless cloning and assembly. (B) The pGp_Tn*7* plasmids were conjugated into strain YS1646 containing a temperature-sensitive plasmid encoding the Tn*7* transposase system. The sequence between the Tn*7* ends containing a chloramphenicol resistance cassette and either rbdA or rbdB sequences was inserted into the YS1646 genome at the *att*Tn*7* site, adjacent to the *glmS* gene. After successful chromosomal integration and loss of both plasmids, Flp-FRT-mediated recombination was used to remove the chloramphenicol resistance cassette. (C) The parent strain YS1646, rbdA, and rbdB expressing YS1646 were individually grown in LB medium overnight. Six hundred seventy nanograms of protein were loaded. A Western blot was performed with a positive control (rrbdA/rrbdB) (7 and 50 ng) produced in *E. coli*. The film was exposed for 15 min (rbdA) and 30 s (rbdB). (D) Parent strain YS1646, rbdA, and rbdB expressing YS1646 were grown in LB, and the $OD_{600}$ was measured every 30 min for 24 h (*n* = 4, one repeat). Data are presented as the mean and standard deviation (SD).

have detectable amounts of antigen following culture. None of the strains had detectable antigen expression 1 h or 24 h after infection of murine macrophages (data not shown). To ensure the fitness of our vaccine strains post-integration, we assessed growth kinetics in LB over 24 h. Only one of the strains (i.e., frr_SspH1_rbdA) had a slower growth rate after chromosomal integration, compared to the parent strain YS1646 (~80%) (Fig. 1D). Because of their ability to drive detectable levels of recombinant antigen expression *in vitro*, the frr_SspH1_rbdA (YS1646::rbdA) and pagC_SspH1_rbdB (YS1646::rbdB) strains were retained and further used for immunogenicity and efficacy testing in mice.

## YS1646::rbdA and YS1646::rbdB elicit high IgG titers when delivered in combination with recombinant rbdA and rbdB delivered intramuscularly

Mice were vaccinated using a multimodal strategy, three doses of the antigen-expressing YS1646 orally on days 0, 2, and 4. On day 0, a dose of recombinant antigen (rrbdA/rrbdB) was also delivered intramuscularly. Delivery of both rrbdA and rrbdB IM with oral delivery of YS1646 not expressing any recombinant antigens served as a positive control, as several groups have shown that the IM vaccination using the receptor binding domains of the *C. difficile* toxins is immunogenic and provides a high level of protection in rodents (35, 41, 42). As expected, IM administration of rrbdA, rrbdB, or both elicited significant antigen-specific IgG titers in the serum of mice 4 weeks after vaccination (Fig. 2A and B). There was a trend toward increased IgG titers in groups that received both IM and oral vaccines compared to the IM-only positive controls, but these differences did not reach statistical significance.

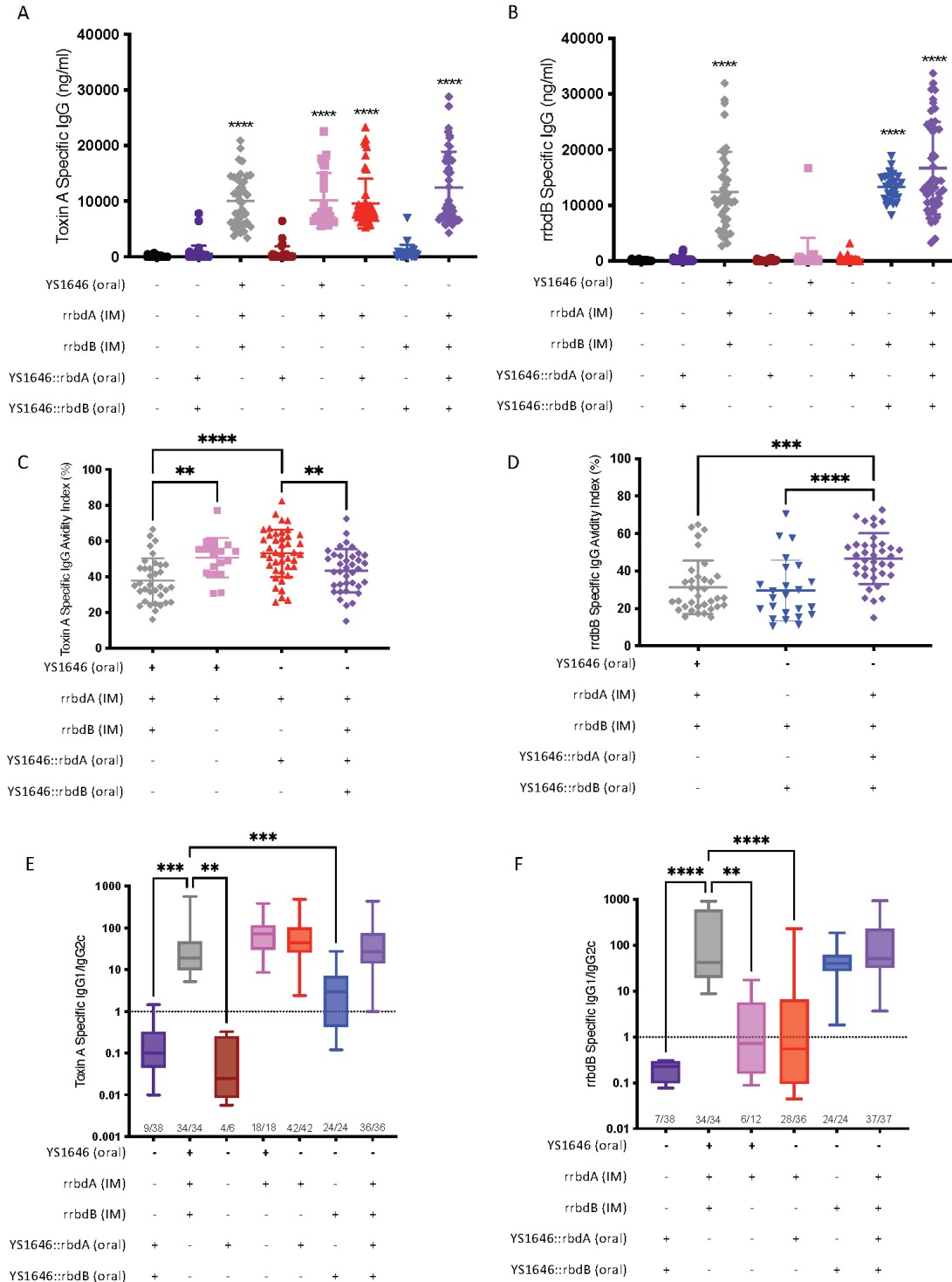

**FIG 2** Multimodal vaccination with YS1646 derivatives expressing rbdA or rbdB generates systemic IgG antibodies. Mice were vaccinated with 3 µg of recombinant antigen (rrbdA/rrbdB) intramuscularly on day 0, with three doses of $1 \times 10^8$ cfu of YS1646 delivered orally on days 0, 2, and 4. Four weeks after vaccination, serum was collected and IgG titers were determined by enzyme-linked immunosorbent assay (ELISA). (A) Toxin A-specific IgG titers are shown as the mean and SD; a multiple-comparison test was used to compare all groups to the PBS group ($n = 25$–$53$, two to six repeats). (B) rrbdB-specific IgG titers are shown the same as A ($n = 25$–$47$, two to four repeats). (C) Toxin A-specific IgG avidity index was determined by (antigen-specific IgG concentration remaining after incubation in 6 M urea)/(total IgG concentration) × 100% . Only groups that consistently had high IgG titers against toxin A were tested for avidity. Data are shown as the mean and SD, with a multiple-comparison test comparing all groups to the rrbdA/B + YS1646 group ($n = 18$–$42$, three to five repeats). (D) rrbdB-specific IgG avidity index was determined the same as in C ($n = 24$–$37$, two to three repeats). (E) Toxin A-specific IgG1/IgG2c ratio was determined

**FIG 2** (Continued)

by (antigen-specific IgG1 titers)/(antigen specific IgG2c titers). A titer below detection was set to 48.75 ng/mL, half of the level of detection. Only mice with detectable titers of at least one of IgG1 or IgG2c were included; the number of mice included per group is indicated above the x axis. The median is displayed as a line, and the whiskers represent the minimum and maximum values. ($n$ = 6–42, two to five repeats). (F) rrbdB-specific IgG1/IgG2c ratio was determined the same as in E ($n$ = 12–38, two to three repeats). All panels were analyzed using the Kruskal-Wallis test and Dunn's multiple-comparison test. $P$ values without a bracket are in comparison to the PBS control group. **$P$ < 0.01; ***$P$ < 0.001; ****$P$ < 0.0001.

To further characterize the IgG elicited by vaccination, we examined the strength of antibody binding (avidity) and subtypes of the systemic IgG in our vaccinated groups compared to the positive control. TcdA-specific IgG avidity was significantly increased in the groups vaccinated only against TcdA, whether they received IM only or IM + oral vaccination (Fig. 2C). IM + oral vaccination against both toxins significantly increased the rrbdB-specific IgG avidity compared to the positive control and vaccination IM + oral against TcdB (Fig. 2D). Mice that received either or both antigens IM generated an IgG1-dominated response, with very low, if any, IgG2c titers (Fig. 2E and F). Mice vaccinated against TcdA only generated some cross-reactive IgG1 and IgG2c antibodies directed against rbdB, with a skewing toward IgG2c. While oral vaccination alone generated IgG titers only in a proportion of the mice, the IgG antibodies were skewed to an IgG2c-dominated response (Fig. 2E and F).

## Vaccination against TcdA using a multimodal strategy, including rbdA expressing YS1646, generates a detectable mucosal response

Since the antigen-expressing YS1646 targets the gut mucosa, the mucosal response to vaccination in the mesenteric lymph nodes (mLNs) and Peyer's patches (PPs) was evaluated in mice vaccinated with YS1646::rbdA orally, rrbdA IM, or both. mLN and PP were collected 32 days after vaccination, and the supernatant from cells stimulated with rrbdA for 72 h was tested for the presence of 16 cytokines and chemokines. Interleukin (IL)-1α secretion in the mLN was significantly increased after IM rrbdA vaccination, with a trend toward increased secretion in the IM + oral vaccinated mice compared to the phosphate buffered saline (PBS) control (Fig. 3A). IL-5 secretion in the mLN was significantly increased after IM + oral vaccination. IL-6, interferon (IFN)γ, IL-17, and granulocyte-macrophage colony-stimulating factor (GM-CSF) had a non-significant increase in secretion in IM and IM + oral vaccinated mice, compared to the PBS and oral-only mice (Fig. S1A). In the PPs, oral vaccination increased GM-CSF secretion significantly (Fig. 3A). Cells from the PPs of mice vaccinated oral or IM alone also had non-significant increases in IL-6 and IFNγ secretion (Fig. S1B). A screening for IgA⁺ plasma cells (PCs) in the mLNs and PPs was performed using ELISpot. Although there was an increase in IgA⁺ PCs in the mLNs of mice vaccinated IM + oral compared to the PBS control (Fig. 3B), this difference did not reach statistical significance ($P$ = 0.0979). There was no significant difference in the number of IgA⁺ in the PPs of vaccinated mice (data not shown).

Intestinal IgA production elicited by vaccination was evaluated in mice vaccinated IM, IM + oral, or oral only against either rbdA, rbdB, or both. The small intestine was collected 5 weeks after vaccination. Although IM vaccination against both toxins failed to elicit antigen-specific IgA titers compared to the PBS control, vaccination with rrbdA IM alone elicited low but detectable levels of TcdA-specific IgA (Fig. 3C). In contrast, oral vaccination against both rbdA and rbdB with or without the IM dose induced a statistically significant increase in antigen-specific IgA. Unexpectedly, IM + oral vaccination against rbdB generated what appear to be cross-reactive antibodies that recognized TcdA. Oral vaccination with either rbdA or rbdB also tended to increase toxin-specific IgA titers in the intestine, but these differences did not reach statistical significance with the relatively small numbers of animals in each group.

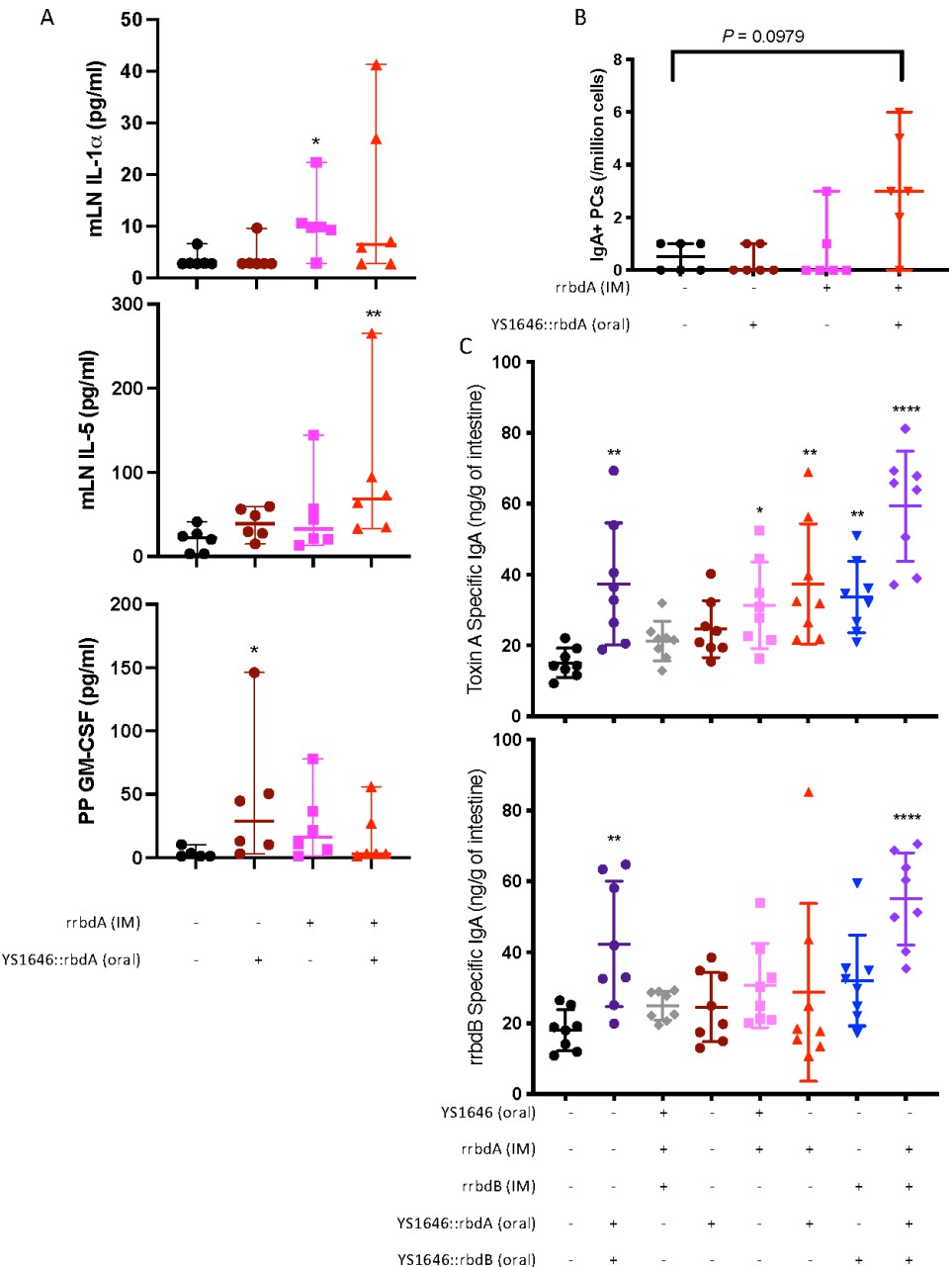

**FIG 3** Oral delivery of rbdA by YS1646 elicits a mucosal immune response. Mice were vaccinated with 3 µg of recombinant antigen (rrbdA and/or rrbdB) intramuscularly on day 0, with three doses of $1 \times 10^8$ cfu of YS1646 given orally on days 0, 2, and 4. Thirty-two days after vaccination, the mesenteric lymph nodes (mLNs) and Peyer's patches (PPs) were collected, and cells were isolated. (A) Cells were stimulated for 72 h with rrbdA, and the supernatant was collected and examined by ELISA to evaluate cytokine and chemokine secretion ($n = 5$–6, two repeats). (B) IgA$^+$ plasma cells were detected by ELISpot ($n = 6$, two repeats). Data are shown as the median and 95% confidence interval. (C) Five weeks after vaccination, the small intestines were collected, and IgA titers were determined by ELISA ($n = 8$, one repeat). Data are presented as the mean and SD. All panels were analyzed using a Kruskal-Wallis test and Dunn's multiple-comparison test to compare all groups to the PBS control. All $P$ values are in comparison to the PBS control group. *$P < 0.05$; **$P < 0.01$; ****$P < 0.0001$.

## Vaccination against both toxins or at high doses against TcdA provides protection to mice from *Clostridioides difficile* challenge

Mice were challenged with *Clostridioides difficile* 5 weeks after vaccination. Of the PBS inoculated control mice, 33% survived the infection. Oral vaccination against TcdA

and IM + oral against TcdB did not significantly protect mice from challenge (Fig. 4A). Mice that were vaccinated IM or IM + oral against TcdA had higher survival than the PBS inoculated mice, although it did not reach statistical significance (80% and 69%, respectively). IM vaccination against both toxins provided significant protection (Fig. 4B). Mice who received the positive control (rrbdA + rrbdB IM) had 100% survival. Mice who had IM + oral vaccination against both toxins had 94% survival. Oral vaccination alone did not protect mice from challenge. In fact, mice that received oral vaccination against both toxins succumbed to infection earlier, with more severe symptoms than the PBS inoculated mice (Fig. 4C). Mice vaccinated against both toxins IM or IM + oral had the least severe symptoms compared to all groups. There appeared to be a pronounced cage effect in some groups. Mice vaccinated against TcdA, either IM or IM + oral, in one cage had no severe symptoms, but in one to three of the other cages, most mice experienced severe symptoms.

TcdA is more toxigenic in mice than TcdB (43). Based on this and the non-significant protection effect that we observed in our model with mice vaccinated against TcdA, we investigated the protection from challenge after higher vaccination doses with rbdA. Both the IM dose (3–10 μg) and the oral dose of YS1646::rbdA ($1 \times 10^8$ cfu/mouse to $1 \times 10^9$ cfu/mouse) were increased. Mice were vaccinated IM + oral or oral only against TcdA. Five weeks after vaccination, mice were challenged with *C. difficile*, and both groups had 100% protection (Fig. 4D). Both groups of mice experienced almost no severe symptoms (Fig. 4E). Only one mouse who received oral only vaccination experienced a symptom score of 10/20 at one timepoint.

Challenge elicited TcdA-specific IgG titers in most surviving mice, with mice that received rrbdA IM tending to have higher titers than the PBS control (Fig. S2A). No rrbdB-specific IgG antibodies were observed in the PBS inoculated mice after challenge, but mice that received rrbdB IM had increased titers compared to the PBS inoculated group. The rrbdB-specific IgG titers were boosted by challenge in mice vaccinated IM + oral against both toxins. The IgG antibodies in all survivors, except mice that received oral vaccination only, were IgG1 skewed (Fig. S2B). Mice that received rrbdA IM had a trend of increased TcdA-specific IgG avidity compared to the PBS inoculated control mice (Fig. S2C). There were no significant differences in rrbdB specific IgG avidity in survivors (Fig. S2D).

IgA titers in the small intestine of surviving mice were examined 3 weeks after challenge. Mice vaccinated IM and IM + oral against both toxins and IM + oral against TcdA had higher TcdA-specific IgA titers compared to PBS mice (Fig. S3). All vaccination strategies, including those targeting TcdA alone, tended to increase rrbdB-specific IgA titers, compared to the surviving PBS inoculated mice.

## DISCUSSION

Our primary goal in this work was to develop stable, recombinant antigen-expressing *Salmonella* Typhimurium vaccine candidates for *C. difficile*. Previous work had established proof of concept that YS1646 expressing repeated motifs of the C-terminal receptor binding domains of either TcdA or TcdB from recombinant plasmids was capable of protecting mice from *Clostridioides difficile* challenge (35). Due to mobility of these plasmids and the presence of an antibiotic resistance cassette, these initial plasmid-containing strains were obviously not suitable for use in humans. While there are several methods for maintaining plasmids in bacteria without the reliance on antibiotic resistance, we chose stable chromosomal integration (29). This permitted us to generate strains with a consistent number of gene copies: initially only one (40). Although this strategy likely reduced expression of the targeted antigens, we tried to mitigate the risk by designing inserts with at least one constitutively active promoter ($P_{frr}$) (36). However, our design still resulted in weak antigen expression. Only two of the strains containing chromosomally integrated recombinant DNA produced levels of heterologous antigens that were detected by Western blot (frr_SspH1_rbdA and pagC_SspH1_rbdB), but both appeared to tolerate the presence of the foreign sequence well, as they had minimal

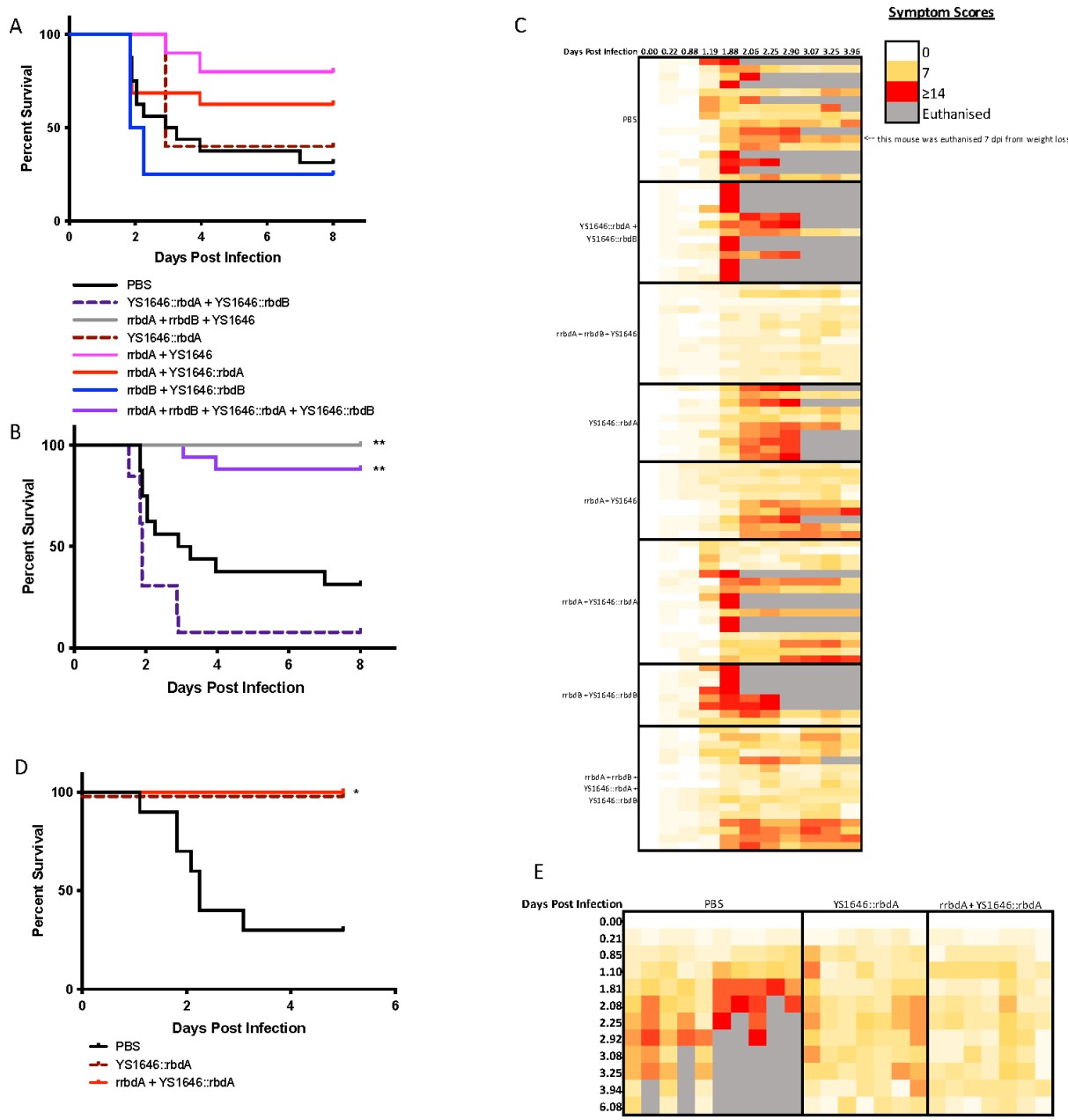

**FIG 4** High-dose vaccination against TcdA provides significant protection against *C. difficile* challenge. (A) Mice from Jackson Laboratories were vaccinated with 3 µg of recombinant antigen (rrbdA/rrbdB) intramuscularly on day 0, with three doses of $1 \times 10^8$ cfu of YS1646 delivered orally on days 0, 2, and 4. At 5 weeks after vaccination, mice were orally challenged with 1,370–2,500 cfu of freshly cultured *C. difficile*. Mouse symptoms were scored one to three times daily by an observer blind to the treatment received. Mice that received a score of 14/20 or above or had over 20% weight loss from their starting weight were at the humane endpoint and were euthanized. Survival of the groups vaccinated against TcdA or TcdB alone is shown (*n* = 8–16, one to two repeats). (B) Survival of the groups vaccinated against both TcdA and TcdB is shown (*n* = 13–16, two repeats). (C) Symptom scores for all groups are shown (*n* = 8–16, one to two repeats). (D) Mice from Charles River Laboratories were vaccinated with a high dose of vaccine, 10 µg of recombinant antigen (rrbdA) intramuscularly on day 0, with three doses of $1 \times 10^9$ cfu of YS1646::rbdA delivered orally on days 0, 2, and 4. At 5 weeks after vaccination, mice were challenged with $1.18 \times 10^7$ cfu of freshly cultured *C. difficile*. Survival is shown (*n* = 7–10, one repeat). (E) Symptom scores of mice that received a high dose of vaccine are shown (*n* = 7–10, one repeat). For all survival curves, the log-rank (Mantel-Cox) test was used to compare all groups to the PBS control group. Correction of the *P* value for multiple comparisons was done using the Bonferroni method. *$P < 0.025$; **$P < 0.00125$.

changes in fitness. These two strains were retained for further studies in mice. Ongoing work with these strains includes in-depth characterization of the stability of the insertions at the attTn7 site through serial passage.

There are several limitations to the mouse models used in this study. The main limitation in our vaccination model is that wild-type *S*. Typhimurium is a mouse pathogen that causes typhoid-like disease in mice (44). It systemically infects mice, with high bacterial burden in the spleen, liver, and gall bladder (45, 46). However, in healthy humans, *S*. Typhimurium is restricted to the gastrointestinal tract and elicits strong mucosal responses (47, 48). We have observed that YS1646 is capable of colonizing the spleen and liver for up to 3 weeks after vaccination (unpublished data). The immune response to *S*. Typhimurium is tissue specific, which is why we focused on responses elicited in the mesenteric lymph nodes, Peyer's patches, and the small intestine. One of the main limitations in the mouse model of CDI is that mice are more sensitive to TcdA than TcdB, while the opposite is believed to be true in humans (43). Syrian hamsters are a very commonly used model for CDI (49). However, their extreme sensitivity to CDI does not mimic infection conditions in humans (50). Mice require antibiotic treatment, albeit a very robust course of antibiotics, to become susceptible to infection, thus better modeling the disease in humans.

The positive control used in this study was IM delivery of one or both recombinant antigens (rrbdA and rrbdB) and oral administration of unmanipulated YS1646. This control generated high IgG titers with a sharply skewed IgG1 profile. As expected, based on our previous work and that of others, this vaccine provided good protection against *C. difficile* challenge (35, 41, 42). This positive control strategy is similar to several of the *C. difficile* vaccine candidates that have entered clinical trials (19–22). Although repeated doses of these candidates administered IM with adjuvants over several months generated strong serum IgG titers in humans, the two vaccines that advanced to efficacy trials failed to meet their primary endpoint of protection against primary CDI (19–21, 23, 24). The field trial of Sanofi's aluminum-adjuvanted, formalin-inactivated whole toxin vaccine (*Cdiffense*: NTC01887912) was terminated at the interim analysis, when the vaccine efficacy was determined to be −5.2% (95% CI, −104.1 to 43.5) (23). More recently, Pfizer reported disappointing results for their aluminum hydroxide-adjuvanted, genetically detoxified toxin candidate vaccine (NTC03090191) (24). In this multi-year study of >17,000 subjects, three doses of the vaccine were only 31% effective at preventing primary CDI (24). The failure of these candidates, despite the induction of high systemic IgG titers, suggests that the possible correlates of protection for CDI should be re-considered in designing the next generation of vaccines. Recently, Cook et al. demonstrated that TcdB-specific CD4[+] T cells in the blood have a stronger negative correlation than humoral responses with disease severity (51). Fecal TcdA-specific IgA titers have been associated with a lower risk of recurrence (52). In a small study, Johal et al. observed a decrease in IgA[+] cells in colonic biopsies of patients with CDI that appeared to correlate with disease severity (53). As a result, the apparent "superiority" of protection provided by the IM-only vaccination compared to oral-only or multimodal vaccination in mice in some of our experiments is unlikely to predict superior efficacy of this traditional approach in the clinic.

We have observed that YS1646-delivered antigen at the intestinal mucosa elicits several different responses to vaccination compared to the IM positive control. Our plasmid-based strains elicited an IgA response in the intestine, which is sustained for 6 months (unpublished data) (35). With our chromosomally integrated antigen expression system, we demonstrated that the mLN of mice that received multimodal vaccination had increased IL-5 production after stimulation with rrbdA, which may account for the increase in TcdA-specific IgA[+] plasma cells and in turn the increased antigen-specific IgA titers observed in the small intestine (54). The differences in expression of the cytokines and chemokines measured at the mLN and PPs after vaccination were subtle and sometimes driven by increased expression in only some of the mice in each group (Fig. S1). Nonetheless, the cytokines that tended to have increased secretion were interesting in the context of a response to *C. difficile*, including IL-1α, IL-6, and IFNγ that are linked to Th1 responses. IFNγ-producing innate lymphoid cells play a role in early resistance to CDI, suggesting that a vaccine that generates a Th1 response would be helpful in

early control of the pathogen (55). GM-CSF has been shown to play a role in neutrophil influx in CDI but does not contribute to pathogen clearance (56). Due to its role in gut inflammation, it is not surprising to see an increase in its production after a *Salmonella* infection (57). Finally, IL-17 may play a role in protection against CDI. Cook et al. found that TcdB-specific Th17 CD4[+] T cells were associated with decreased risk of recurrence (51). Chen et al. recently demonstrated that IL-17 production by γδ T cells is vital in neonatal resistance to CDI (58). While the increase in the production of these cytokines was not significant in our vaccine groups, it is positive that the cytokines that had a small signal would be helpful in the context of CDI.

IgG1 and IgG2c production gives us an indication about the Th skewing of the immune response generated by our vaccines. IgG1 is produced during a Th2 skewed response, while a Th1 response elicits IgG2c antibodies (59). We observed a dichotomy in the production of IgG1 and IgG2c by vaccinated mice. Most mice that produced high IgG titers produced high IgG1 titers and very low IgG2c titers. In mice with very low IgG titers, the antibodies produced were mainly IgG2c. This is expected as aluminum hydroxide gel, the adjuvant in our IM delivered vaccine, is Th2 skewing, while intracellular infections, such as *S.* Typhimurium, elicit Th1 responses (60). Correlates of protection in CDI were discussed above, but as an extracellular pathogen in the intestine, both Th1 and Th2 responses may be needed to clear the infection. With this in mind, multimodal vaccination may be a vital strategy to elicit a mixed Th1/Th2 response.

While generally consistent with our prior findings using plasmid-based expression in YS1646, the current work has several limitations (35). For example, protection with a lower dose vaccination and challenge model in the current work was dependent on exposure of the mice to both toxins unlike the plasmid-based model in which multimodal vaccination against either toxin was protective (35). It is worth noting that part of our motivation for the changes in the model between our prior studies and the current experiments was an unexpected difference in the apparent sensitivity of the C57BL/6J mice acquired from either Charles River Laboratories (relatively resistant) or Jackson Laboratories (relatively sensitive). We hypothesize that this observation may be attributable to differences between the mice bred in these two facilities in the sensitivity of their gut microbiota to the antibiotic "cocktail" used and possible exposures to non-toxigenic *C. difficile* strains (61). Since we only examined the mucosal response in the low-dose model, it is possible that higher dose antigen-expressing YS1646 may be better at eliciting responses at the gut mucosa. In addition, our studies examining the mucosal responses include relatively small numbers of mice, and some of the trends observed were driven by only some of the animals in each group. Interestingly, we observed some evidence of antigen interference in the mice vaccinated against both toxins in this study. TcdA-specific IgG avidity was higher in mice vaccinated against TcdA alone, compared to mice vaccinated against both toxins. While bacterial competition of our vaccine strains could be contributing to this issue, as YS1646::rbdB does grow faster than YS1646::rbdA in LB, the trend is also present in mice vaccinated IM only, suggesting that the difference in response is antigenic in nature. Finally, this study did not examine either T cell responses or the durability of responses elicited by vaccination. T cells are thought to play a role in maintaining long-term responses to vaccination. Future studies by our group will examine these aspects of the immune response, as the necessity for a vaccine to elicit highly durable responses has been highlighted by the decreased vaccine effectiveness over time observed in Pfizer's phase 3 clinical trial (24).

The ultimate goal of this program is to develop a novel vaccine candidate against *C. difficile* that is safe and can provide both rapid and durable protection in vulnerable human populations. One of the main benefits of repurposing the YS1646 strain is its documented safety profile in mice, pigs, non-human primates, and humans (30, 31, 34). The combination of this living vaccine vector in our multimodality regimen elicits a mucosal response that may be vital for protection from CDI. While the multimodality approach adds complications to the regulatory process (i.e., more components for safety testing) and packaging, the application of this strategy would be relatively simple in

a clinical practice. In a single visit, both the IM dose and the initial oral dose could be administered, with the remaining oral doses taken at home, as is currently done for the oral *S.* Typhi vaccine (62). Although the oral only strategy was generally not as effective as combined IM + oral vaccination, it is possible that oral vaccination alone with antigen-expressing YS1646 may be sufficient to provide protection in humans. To further complicate initial clinical trials, it is possible that vaccination against TcdB may be sufficient to provide protection. However, the vaccines that have undergone phase II/III clinical trials have all targeted both TcdA and TcdB, so a multi-antigen strategy may be necessary (20, 23). As one or more of the YS1646 candidates move forward into clinical trials, we will test both administration strategies.

## MATERIALS AND METHODS

### Bacterial strains and growth conditions

*Salmonella enterica* Typhimurium YS1646 (Δ*msbB2* Δ*purI* Δ*Suwwan xyl* negative; ATCC 202165; ATCC, Manassas, VA) was obtained from Cedarlane Labs (Burlington, ON, Canada). *Escherichia coli* DH5α (Thermo Fisher Scientific, Eugene, OR) was used to produce recombinant Tn*7* plasmids. *E. coli* MGN-617 was used for conjugation with YS1646 (63). Plasmids were introduced into YS1646 either by conjugation or by electroporation (20 ng of plasmid at 1.8 kV, 200 Ω and 25 µF; ECM 399 Electroporation System, BTX, Holliston, MA, USA) Plasmids were introduced into *E. coli* by heat shock. *Salmonella* Typhimurium and *E. coli* were cultured in LB, with the following antibiotics when necessary to maintain plasmids: 100 µg/mL ampicillin, 50 µg/mL kanamycin, 30 µg/mL chloramphenicol, and if necessary 50 µg/mL diaminopilmelic acid (DAP).

For the growth curves, cultures of wild-type YS1646 and chromosomally integrated constructs were grown overnight at 37°C. The next day, the cultures were diluted 1:100 in LB and plated in quadruplicates ($n = 4$) on a 100-well Bioscreen C honeycomb microplate (Growth Curves USA, Piscataway, NJ, USA). The Bioscreen C plate reader measured the optical density of the cultures at a wavelength of 600 nm every 30 min over 24 h with a 30-s shaking period prior to each reading.

*Clostridioides difficile* strain VPI 10463 (ATCC 43255) was obtained from Cedarlane Labs and was used for challenge experiments. Cells were maintained in meat broth (Sigma-Aldrich, St. Louis, MO) containing 0.1% (wt/vol) L-cysteine (Sigma-Aldrich) in an anaerobic jar. For colony counts, *C. difficile*-containing medium was serially diluted and streaked onto brain-heart infusion-supplemented plates (BD Biosciences, Mississauga, ON, Canada) containing 0.1% (wt/vol) L-cysteine. The bacteria were left to grow on plates at 37°C in an anaerobic jar for 24 h.

### Chromosomal integration

The chloramphenicol-resistant Tn*7* plasmid backbone was originally developed by Crépin et al. (40). The pGP-Tn*7*-Cm plasmid backbone was digested using EcoRI and KpnI, and the promoter, secretory signal, and antigen sequence were inserted using uni seamless cloning and assembly (pEASY kit, TransGen Biotech, Beijing, China). All primers used in the construction of Tn*7* plasmids are described in Table S1. Most promoter, secretory signal, and antigen sequences were originally made by Winter et al. (35). The *frr* promoter was obtained from YS1646 using PCR. After assembly of the Tn*7* plasmids, they were transformed into DAP⁻ *E. coli* κ12c617.

The temperature-sensitive pSTNSK plasmid, designed by Crépin et al., containing the Tn*7* transposase system and a kanamycin resistance cassette, was transformed into YS1646 (40). The transformed *E. coli* κ12c617 and *S.* Typhimurium were incubated together in 10 µL of LB at 30°C for 5 h. They were then plated on LB agar with kanamycin and chloramphenicol but no additional DAP. Individual colonies of YS1646 were selected and grown at 42°C to lose the pSTNSK plasmid, and PCR was performed to confirm chromosomal integration.

Chromosomally integrated YS1646 was transformed with the temperature-sensitive pCP20 plasmid (64), containing an ampicillin resistance cassette, a chloramphenicol resistance cassette, and the recombinase flippase (Flp). The chloramphenicol resistance gene, *cat*, which was integrated at the *att*Tn*7* site of the YS1646 genome, was then removed by Flp-FRT recombination. Successful removal of the chloramphenicol resistance cassette in the genome was confirmed both by determining antibiotic susceptibility to chloramphenicol and by PCR.

## Recombinant rbdA and rbdB expression

Protein expression and purification of the recombinant $TcdA_{1820-2710}$ (rrbdA) and $TcdB_{1821-2366}$ (rrbdB) were accomplished using the pET-28b plasmid (Novagen, Millipore Sigma, Burlington, MA) with an isopropyl-β-D-1-thiogalactopyranoside (IPTG)-inducible promoter and kanamycin resistance gene. A 6× His tag and stop codon were added at the 3′ end. The expression vector was transformed into *E. coli* BLR(DE3) cells (Novagen, Millipore Sigma) by heat shock. Transformed bacteria were grown in a 37°C shaking incubator with 30 µg/mL of kanamycin (Wisent), until the optical density (absorbance) at 600 nm ($OD_{600}$) reached 0.5 to 0.6. IPTG (Invitrogen, Carlsbad, CA) was then added to a final concentration of 1 mM, and expression was induced overnight. Expression of rrbdA was done at 37°C, while expression of rrbdB was performed at 30°C. Cells were pelleted by centrifugation at $3,000 \times g$ for 30 min at 4°C. Cells were lysed, and the lysate was collected and purified using a denaturing protocol and Ni-nitrilotriacetic acid (NTA) affinity chromatography (Ni-NTA Superflow; Qiagen, Venlo, Limburg, Netherlands). The eluate was analyzed by Coomassie blue staining of polyacrylamide gels and Western blotting using a monoclonal antibody directed against the His tag (Sigma-Aldrich).

## Western blotting

For antigen expression *in vitro*, the transformed YS1646 strains were grown overnight in LB at 37°C in 0% $CO_2$. Overnight cultures were centrifuged at $21,130 \times g$ for 10 min and resuspended in PBS. Six hundred seventy nanograms of each sample and 7 ng and 50 ng of positive controls were then mixed in with NuPAGE lithium dodecyl sulfate sample buffer (Invitrogen) according to the manufacturer's instructions. All samples were heated for 10 min at 100°C and then cooled on ice. Proteins were separated on a 4% to 20% Bis-Tris protein gel (Invitrogen) and transferred to nitrocellulose membranes using a Trans-Blot Turbo RTA mini nitrocellulose transfer kit (Bio-Rad, Hercules, CA). For detection of $TcdA_{5458-8130}$ and $TcdB_{5461-7080}$, the membranes were incubated first with anti-toxin A chicken IgY (1:5,000; Abnova, Taipei, Taiwan) and anti-toxin B chicken IgY (1:10,000; Abnova) antibodies, respectively, followed by goat anti-chicken-IgY IgG conjugated to horseradish peroxidase (HRP; 1:10,000; Thermo Fisher Scientific). Immunoreactive bands were visualized using the SuperSignal West Pico Plus chemiluminescent substrate (Thermo Fisher Scientific) and autoradiography film (Denville Scientific, Holliston, MA).

## Mice

Six- to eight-week-old female C57BL/6J mice were obtained from Jackson Laboratory (Bar Harbor, Maine, USA) or Charles River Laboratories (Montreal, QC, Canada) and were kept under pathogen-free conditions in the Animal Resource Division at the McGill University Health Center Research Institute. All animal procedures were approved by the Animal Care Committee of McGill University and performed in accordance with the guidelines of the Canadian Council on Animal Care.

## Vaccination

For oral vaccinations, mice were gavaged with $1 \times 10^8$ cfu of the YS1646 strains in 0.2 mL of PBS (days 0, 2, and 4). When both strains were given, $5 \times 10^7$ cfu of each strain was used, for a total of $1 \times 10^8$ cfu of YS1646 given in 0.2 mL of PBS. IM injections contained 3 µg of recombinant protein and 250 µg of aluminum hydroxide gel (alum; Alhydrogel;

Brenntag BioSector A/S, Frederikssund, Denmark) in 50 µL, which was administered into the gastrocnemius muscle using a 28-gauge needle. For the high-dose study, mice were gavaged with $1 \times 10^9$ cfu of the YS1646 strains, and IM injections contained 10 µg of recombinant protein.

## Blood and intestine sampling

Baseline and pre-infection serum samples were collected from the lateral saphenous vein prior to all other study procedures using Microtainer serum separator tubes (Sarstedt, Nümbrecht, Germany). Serum samples were also collected from the mice at the end of the study by cardiac puncture after isoflurane-$CO_2$ euthanasia. Serum separation was performed according to the manufacturer's instructions, and aliquots were stored at −20°C until they were used. At study termination, 10 cm of the small intestine, starting at the stomach, was collected. Intestinal contents were removed, and the tissue was weighed and stored in a protease inhibitor cocktail (catalog number P8340; Sigma-Aldrich) at a 1:5 (wt/vol) dilution on ice until it was processed. The tissue was homogenized (Homogenizer 150; Fisher Scientific, Ottawa, ON, Canada) and centrifuged at $2,500 \times g$ at 4°C for 30 min, and the supernatant was collected. Supernatants were stored at −80°C until they were analyzed by enzyme-linked immunosorbent assay (ELISA). For post challenge data, samples were collected from survivors at 3 weeks after infection.

## Mesenteric lymph node and Peyer's patches sampling

mLNs and PPs were excised and collected from mice 30 and 32 days after vaccination, in Gibco Roswell Park Memorial Institute (RPMI) 1640 medium supplemented with 1 mM penicillin/streptomycin, 10 mM HEPES, 1× MEM non-essential amino acids, 1 mM sodium pyruvate, 1 mM L-glutamine (all Wisent products), and β-mercaptoethanol (Sigma-Aldrich) (cRPMI) and 2% fetal bovine serum (FBS) and kept on ice. All mLNs and PPs from individual mice were transferred into 1 mL of digestion buffer [1 mg/mL of Collagenase D (Sigma-Aldrich) and 0.1 mg/mL of DNase I (Sigma-Aldrich) in RPMI 1640 + 2% FBS] and cut open. They were then incubated at 37°C, shaking at 220 rpm for 40 min. The media and cells were passed through a 70-µm cell strainer (BD Biosciences) and washed with cRPMI + 2% FBS three times. Cells were resuspended in cRPMI + 10% FBS.

## *Clostridioides difficile* challenge

*C. difficile* challenge experiments were performed essentially as described previously (35, 65–67). Briefly, mice were pre-adapted to acidic water by adding acetic acid at a concentration of 2.15 µL/mL (vol/vol) to their drinking water 1 week prior to antibiotic treatments. At 6 days prior to infection, an antibiotic cocktail that included metronidazole (0.215 µg/mL; Sigma-Aldrich), gentamicin (0.035 µg/mL; Wisent), vancomycin (0.045 µg/mL; Sigma-Aldrich), kanamycin (0.400 µg/mL; Wisent), and colistin (0.042 µg/mL; Sigma-Aldrich) was added to the drinking water. After 3 days, regular water was returned, and 24 h prior to infection, mice received clindamycin (32 mg/kg of body weight; Sigma-Aldrich) intraperitoneally in 0.2 mL of PBS using a 28-gauge needle. Fresh *C. difficile* cultures were used in our challenge model so that the dose used was estimated on the day of infection based on $OD_{600}$ values and the precise inoculum was calculated 24 h later. This procedure led to the use of different *C. difficile* doses in the two repeat challenge studies performed (1,370 or 2,500 cfu/mouse). In the high-dose study, mice received $1.18 \times 10^7$ cfu/mouse. The challenge dose was delivered by gavage in 0.2 mL of meat broth culture medium. The mice were then monitored and scored one to three times daily for weight loss, activity, posture, coat quality, diarrhea, and eye/nose symptoms (Table S2) (66). A score of 14/20 or above and/or 20% weight loss were considered as a humane endpoint, and mice were euthanized. Any mouse found dead was given a score of 20. Survivors were followed and euthanized approximately 3 weeks after infection.

## Enzyme-linked immunosorbent assay

Whole toxin A (List Biologicals, Campbell, CA) or recombinant rbdB was used to coat U-bottom high-binding 96-well ELISA plates (Greiner Bio-One, Frickenhausen, Germany). A standard curve was generated for each plate using mouse IgG antibodies, mouse IgG1 antibodies, mouse IgG2c antibodies, or mouse IgA antibodies (Sigma-Aldrich). The plates were coated with 50 µL of toxin A (1.0 µg/mL), rrbdB (0.25 µg/mL), or IgG/IgG1/IgG2c/IgA standards overnight at 4°C in 100 mM bicarbonate/carbonate buffer (pH = 9.5). The wells were washed with PBS three times and then blocked with 150 µL of 2% bovine serum albumin (Sigma-Aldrich) in PBS–Tween 20 (0.05%; blocking buffer; Fisher Scientific) for 1 h at 37°C. Serum samples were heat inactivated at 56°C for 30 min before dilution 1:50 in blocking buffer. Intestinal supernatants were not heat inactivated and were added to the plates neat. All sample dilutions, including dilutions for the standard curve, were assayed in duplicate (50 µL/well). The plates were incubated for 1 h at 37°C and then washed four times with PBS. For avidity assays, 100 µL of blocking buffer or 6 M urea was added to samples on the same plate for 15 min. Urea was washed off with PBS four times and then plates were blocked with 150 µL of blocking buffer for 1 h at 37°C. After washing the samples or the blocking buffer off (avidity assay), 75 µL of HRP-conjugated anti-mouse total IgG antibodies (1:20,000 in blocking buffer; Sigma-Aldrich), HRP-conjugated anti-mouse IgG1 antibodies (1:20,000 in blocking buffer; Sigma-Aldrich), HRP-conjugated anti-mouse IgG2c antibodies (1:20,000 in blocking buffer; Sigma-Aldrich), or HRP-conjugated anti-mouse IgA antibodies (1:10,000 in blocking buffer; Sigma-Aldrich) was added. The plates were incubated for 30 min (IgG, IgG1, and IgG2c) or 1 h (IgA) at 37°C. Six washes with PBS were performed before the addition of 100 µL/well of 3,3′,5,5′-tetramethylbenzidine detection substrate (Millipore, Billerica, MA). Reactions were stopped after 15 min with 50 µL/well of 0.5 M $H_2SO_4$. The plates were read at 450 nm on an EL800 microplate reader (BioTek Instruments, Inc., Winooski, VT). The concentration of antigen-specific antibodies in each well (in nanograms per milliliter) was estimated by extrapolation from the standard curve. Avidity index was calculated as (antigen-specific IgG concentration remaining after urea incubation)/(total IgG concentration) × 100%.

## ELISpot

A mouse IgA ELISpot basic kit from Mabtech (Stockholm, Sweden) was used. Briefly, hydrophobic PVDF membrane ELISpot plates (Millipore Sigma) were coated with 100 µL/well of the anti-IgA capture antibody overnight at 4°C. Plates were then blocked with cRPMI for at least 30 min at room temperature. mLN cells in cRPMI were added to the plate at $5 \times 10^5$ cells/well and were incubated for 24 h. After washing, 1 µg/mL of biotinylated TcdA (biotinylated using Thermo Fisher kit, A39257) was incubated for 2 h at room temperature. The plate was washed, and streptavidin-ALP (1:10,000) was added to each well and left to incubate at room temperature for 1 h. BCIP/NBT-plus substrate (Mabtech) was added, and the plate developed for 10 min. Plates were read on a CTL series 3B ImmunoSpot analyzer (Cellular Technology Limited, Ohio) with ImmunoSpot 5.2 analyzer software.

## Cytokine quantification (Quansys)

mLN and PP cells were plated in cRPMI at $2.5 \times 10^6$ cells/well and were stimulated with 2 µg/mL of rrbdA for 72 h. Supernatant was collected and stored at −80°C until further use. The concentrations of 16 cytokines and chemokines (IL-1α, IL-1β, IL-2, IL-3, IL-4, IL-5, IL-6, IL-10, IL-12p70, IL-17, MCP-1/CCL2, IFNγ, TNFα, MIP-1α/CCL3, GM-CSF, and RANTES/CCL5) were determined using the Q-Plex Mouse Cytokine Screen (16-plex) multiplex ELISA following the manufacturer's guidelines (Quansys Biosciences, Utah). Samples were run in singlet.

## Statistical analysis

Statistical analysis was performed using GraphPad Prism (version 9) software. For analysis of antibody titers, a one-way nonparametric Kruskal-Wallis analysis of variance was performed with Dunn's multiple-comparison analysis for comparison of all groups. Statistical significance was considered to have been achieved when $P$ was 0.05. For analysis of survival, the log-rank (Mantel-Cox) test was used to compare all groups to the PBS control group. The Bonferroni method was used to correct for multiple comparisons.

## ACKNOWLEDGMENTS

We would like to thank Annie Beauchamp for assistance with all animal work carried out in this study. We thank Adam Hassan for his help with the growth curve and Western blots and Hilary Hendin for her help with the ELISpot assay. We thank Li Xing for his work in designing the initial promoter, secretory signal, and antigen expression cassettes.

All authors contributed to the study and experimental design, with S.H. and C.M.D. lending their considerable expertise to the bacterial genetic experiments. K.W. and S.H. performed the chromosomal integration. K.W. performed all animal experiments. K.W. and B.J.W. performed the analysis of the data and prepared the manuscript.

## AUTHOR AFFILIATIONS

[1]Department of Microbiology and Immunology, McGill University, Montreal, Québec, Canada

[2]Research Institute of the McGill University Health Centre, Montreal, Québec, Canada

[3]Institut National de Recherche Scientifique–Centre Armand-Frappier Santé Biotechnologie, Laval, Québec, Canada

## AUTHOR ORCIDs

Kaitlin Winter http://orcid.org/0000-0002-2626-1730
Charles M. Dozois http://orcid.org/0000-0003-4832-3936
Brian J. Ward http://orcid.org/0000-0003-3251-958X

## FUNDING

| Funder | Grant(s) | Author(s) |
|---|---|---|
| Gouvernement du Canada \| Canadian Institutes of Health Research (IRSC) | IPR-144157 | Brian J. Ward |
| Aviex Technologies LLC | IPR-144157 | Brian J. Ward |
| Gouvernement du Canada \| Natural Sciences and Engineering Research Council of Canada (NSERC) | | Charles M. Dozois |
| FRQ \| Fonds de Recherche du Québec - Santé (FRQS) | | Kaitlin Winter |
| Faculty of Medicine, McGill University (McGill Faculty of Medicine) | | Kaitlin Winter |

K.W. received studentships from the Faculty of Medicine at McGill University and the Fonds de Recherche en Santé de Québec. This work was funded by a Canadian Institutes of Health Research (CIHR) industry grant with Aviex Technologies LLC (IPR-144157) awarded to B.J.W. and a Natural Sciences and Engineering Research Council (NSERC) Discovery grant to C.M.D.

## AUTHOR CONTRIBUTIONS

Kaitlin Winter, Conceptualization, Data curation, Formal analysis, Methodology, Writing – original draft | Sébastien Houle, Data curation, Methodology, Validation, Writing – review and editing | Charles M. Dozois, Conceptualization, Methodology, Supervision, Writing – review and editing | Brian J. Ward, Conceptualization, Formal analysis, Funding acquisition, Investigation, Methodology, Project administration, Resources, Supervision, Writing – original draft, Writing – review and editing

## DATA AVAILABILITY

The data that support the findings of this study are available from the corresponding author upon reasonable request.

## ADDITIONAL FILES

The following material is available online.

### Supplemental Material

**File S2 (Spectrum03109-22- S0001.txt).** Sequence of YS1646 at the *glmS* site.
**File S3 (Spectrum03109-22- S0002.txt).** Sequence of frr_SspH1_rbdA insert at the *glmS* site.
**File S4 (Spectrum03109-22- S0003.txt).** Sequence of pagC_SspH1_rbdB insert at the *glmS* site.
**Supplemental material (Spectrum03109-22- S0004.pdf).** Tables S1 and S2; Figures S1, S2, and S3.

### Open Peer Review

**PEER REVIEW HISTORY (review-history.pdf).** An accounting of the reviewer comments and feedback.

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
