## [Reviewer comments · Microbiology Spectrum]

Microbiology Spectrum

Multimodal vaccination targeting the receptor binding domains of *Clostridioides difficile* toxins A and B with an attenuated *Salmonella Typhimurium* vector (YS1646) protects mice from lethal challenge

Kaitlin Winter, Sébastien Houle, Charles Dozois, and Brian Ward

Corresponding Author(s): Brian Ward, Research Institute of the McGill University Health Centre

Review Timeline:

Submission Date:	August 9, 2022
Editorial Decision:	October 17, 2022
Revision Received:	November 3, 2022
Editorial Decision:	November 18, 2022
Revision Received:	November 29, 2023
Accepted:	December 6, 2023

Editor: Meera Unnikrishnan

Reviewer(s): The reviewers have opted to remain anonymous.

Transaction Report:

DOI: <https://doi.org/10.1128/spectrum.03109-22>

October 17, 2022

Dr. Brian J Ward
Research Institute of the McGill University Health Centre
1001 Decarie Blvd
Montreal, Quebec
Canada

Re: Spectrum03109-22 (Vaccination with an attenuated *Salmonella* Typhimurium vector (YS1646) with chromosomal expression of *Clostridioides difficile* antigens completely protects mice from lethal challenge)

Dear Dr. Brian J Ward:

Thank you for submitting your manuscript to Microbiology Spectrum.

We have received comments back from two expert reviewers which are included below. Although they find the paper interesting, several experimental details need further clarification. Please submit a revised manuscript addressing the Reviewers' concerns.

Link Not Available

Sincerely,

Meera Unnikrishnan

Journals Department
Reviewer comments:

Reviewer #1 (Comments for the Author):

In this work, authors develop an attenuated *Salmonella* Typhimurium vector with an ectopic copy of *C. difficile* toxin-specific antigens and demonstrate that these antigens provide protection against lethal *C. difficile* challenge in a murine model.

The manuscript is well written however, it can be improved for better clarity to the readers. The study is, in most experiments,

well executed and with adequate rigor. There are several major concerns that need to be concealed.

Among my major concerns:

The title needs to be revised so that it more accurately reflects the major conclusion from this work. The major claim stated in the title is that this vaccination strategy provides "complete" protection to mice against *C. difficile* challenge. This is somehow misleading as authors did not account for colonization of immunized mice. Please correct accordingly.

In general, the abstract is 155 words long, and is extremely general lacking proper description of some key experiments. Expand the types of vaccinations performed, what is meant by multimodal vaccination strategy? Indicate the other vaccination strategies used in the study. Readers should be able to understand the experimental design from the abstract as well.

A major concern with this work is the lack of rigor to confirm that chromosomally integrated chimeras and/or the strain itself does not have off mutations. This is not stated in the methods and if strains were indeed whole genome sequence, the illumine reads should be made available in a public domain.

Another major concern regards toxin expression in attenuated *Salmonella* strains. The western blot shown, barely has detectable immunoreactive bands and lacks proper loading control. That is another protein that can serve the purpose to ensure that authors loaded similar amounts of cell lysate per well. This is a requirement. In addition, proper quantification of chimera production by the different strains needs to be confirmed by ELISA, which was already used in downstream experiments.

Minor concerns:

Line 15: specify how was the delivery.

Line 16-18: not clear what is meant by cassettes where stably-integrated. Where? how where they regulated in *Salmonella*? Please provide more details.

Lines 36-37: these should also be in the abstract.

Line 37: was oral vaccination a separate treatment than the multimodal? It is difficult to understand as written.

Lines 60-64: readers would appreciate more information regarding these vaccines in Phase 2/3 clinical trials. Antigen employed, mode of delivery, company, etc.

Line 64: what is being meant by high-profile failure? Please re-phrase.

Line 65-68: Include the rationale of neutralizing both toxins here as well.

Lines 89-92: no reference for this statement. Is this part of this work or previous work by the authors?

Lines 100-102: Abbreviation for oral (PO) is confusing; no abbreviation for multimodal vaccination is provided. It is difficult to differentiate between PO and multimodal.

Line 248: Although authors state that *pfr* and *ppagC* are strong promoters, toxin production in the attenuated strain is weak as observed by western blotting.

Fig supplementary 1: The figure legend fails to describe the treatments. What is IM, PO and IM+PO? It is not clear which is the PBS control. Please revise.

Fig supp 2: same comments as in fig S1.

Fig supp 3: same comments as in fig S1.

Reviewer #2 (Comments for the Author):

Authors used *Salmonella* Typhimurium (YS1646) strain to express receptor binding domains (RBDs) of *Clostridioides difficile*. RBDs were integrated into YS1646 to avoid antibiotic marker carried on the expression plasmid. They evaluated the ST constructs in comparison with RBDs separately or in combination as multimodal vaccine candidates in mouse model of *C. difficile* infection (CDI). While data is interesting, it seems not support the claimed advantages of using ST as a component of an effective vaccine against CDI.

1) Need more information in the introduction on the safety of this YS1646 strain, why and how it is safe.

2) Animal survival assays showed combined immunization of ST and RBDs were not better (actually worse) than immunization with RBDs only?

3) No rationale on avidity assays. What does this mean?

4) No resource information on biotinylated TcdA

5) fecal sample (intestinal contents) IgG and IgA should be determined for mucosal vaccination.

5) IgA and IgM in sera should be determined.

- 6) More confusing is the mouse model of CDI. Fig4 A and Fig 4B used different challenge dosages without rationale? I am also confused by FigB (challenged with toxins?) Fig C is not useful.
- 7) sFig 1 is hard to read. Why not use a clearer column figure?
- 8) sFig 2: one group only used 1 mouse!?
- 9) Group size in animal experiment is confusing. 1, 6,....?

Staff Comments:

Preparing Revision Guidelines

Please return the manuscript within 60 days; if you cannot complete the modification within this time period, please contact me. If you do not wish to modify the manuscript and prefer to submit it to another journal, please notify me of your decision immediately so that the manuscript may be formally withdrawn from consideration by Microbiology Spectrum.

Our responses to the Reviewers' comments and suggestions are summarized in a table below with changes in the revised Manuscript indicated where relevant. All simple editorial recommendations have been incorporated into the manuscript; substantive comments are addressed below.

	Comments	Responses
Reviewer #1		
Major Comments		
1.	The title needs to be revised so that it more accurately reflects the major conclusion from this work. The major claim stated in the title is that this vaccination strategy provides "complete" protection to mice against C. difficile challenge. This is somehow misleading as authors did not account for colonization of immunized mice. Please correct accordingly.	The title has been changed to "Vaccination with an attenuated Salmonella Typhimurium vector (YS1646) with chromosomal expression of Clostridioides difficile antigens protects mice from lethal challenge"
2.	In general, the abstract is 155 words long, and is extremely general lacking proper description of some key experiments. Expand the types of vaccinations performed, what is meant by multimodal vaccination strategy? Indicate the other vaccination strategies used in the study. Readers should be able to understand the experimental design from the abstract as well.	The abstract has been modified to address this concern. "Developing a vaccine against Clostridioides difficile is a key strategy to protect the elderly. Two candidate vaccines using a traditional approach of intramuscular (IM) delivery of recombinant antigens targeting C. difficile toxins A (TcdA) and B (TcdB) failed to meet their primary endpoints in large phase 3 trials. To elicit a mucosal response against C. difficile, we repurposed an attenuated strain of Salmonella Typhimurium (YS1646) to deliver the receptor binding domains (rbd) of TcdA and TcdB to the gut-associated lymphoid tissues, to elicit a mucosal response against C. difficile. In this study, YS1646 candidates with either rbdA or rbdB expression cassettes integrated into the bacterial chromosome at the attTn7 site, were generated and used in a short-course multimodal vaccination strategy that combined oral delivery (PO) of the YS1646 candidate(s) on days 0, 2 and 4 and IM delivery of recombinant antigen(s) on day 0. Five weeks after vaccination, mice had high serum IgG titers and increased intestinal antigen-specific IgA titers. Multimodal vaccination increased the IgG avidity compared to the IM only control. In the mesenteric lymph nodes, we observed increased IL-5 secretion, and increased IgA⁺ plasma cells. PO vaccination skewed the IgG response towards IgG2c dominance (vs IgG1 dominance in the IM only group). Both PO alone and multimodal vaccination against TcdA protected mice from lethal C. difficile challenge (100% survival vs 30% in controls). Given the established safety profile of YS1646 we hope to move this vaccine candidate forward into a

		Phase I clinical trial.” (Lines 13-30)
3.	A major concern with this work is the lack of rigor to confirm that chromosomally integrated chimeras and/or the strain itself does not have off mutations. This is not stated in the methods and if strains were indeed whole genome sequence, the illumine reads should be made available in a public domain.	The work to fully-characterize the location and stability of the insertions in the chromosomally-integrated chimeras is on-going and the manuscript has been edited to better reflect our intentions. However, these YS1646 strains are proprietary and will only be made available to other groups under an appropriate confidentiality and/or licensing agreement. This includes the whole genome sequence of the strain. “These two strains were retained for further studies in mice. Ongoing work with these strains includes in depth characterization of the stability of the insertions at the attTn7 site through serial passage and whole genome sequencing to ensure that no off mutations have been introduced during chromosomal integration. ” (Lines 262-265)
4.	Another major concern regards toxin expression in attenuated Salmonella strains. The western blot shown, barely has detectable immunoreactive bands and lacks proper loading control. That is another protein that can serve the purpose to ensure that authors loaded similar amounts of cell lysate per well. This is a requirement. In addition, proper quantification of chimera production by the different strains needs to be confirmed by ELISA, which was already used in downstream experiments.	We knew, from previous work with YS1646 strains bearing different plasmids designed to express the receptor-binding domains (rbd) of C. difficile toxins A or B that there was no correlation between in vivo immunogenicity and in vitro production (ie: axenic culture or in murine RAW264.7 cells) [Winter K et al 2019], As a result, the western blots (WB) used in the current work were intended to be a qualitative screening tool only. Had none of the chromosomally-integrated strains produced the rbd domains detectable by WB, we would have proceeded to in vivo testing with all of the candidates. Since two of the strains were ‘positive’ in the WB screening, these two were chosen to advance. We have modified the text to clarify how the WBs were used in the current study (Lines 136-137): “ In vitro antigen expression in the vaccine candidates was evaluated qualitatively by western blotting.” More detail for the WB assay has been provided as follows (Lines 451-456): “For antigen expression in vitro , the transformed YS1646 strains were grown overnight in LB at 37°C in 0% CO ₂ . 50µL of overnight cultures were centrifuged at 21,130xg for 10 min and resuspended in 100µL PBS. Samples were then mixed in with NuPAGE lithium dodecyl sulfate (LDS) sample buffer (Invitrogen) according to the manufacturer’s instructions. All samples were heated for 10 min at 100°C and then cooled on ice. 15µl of sample was loaded in each lane, proteins were separated on a 4 to 12% Bis-Tris protein gel (Invitrogen)” We do not have a sandwich ELISA optimized in our lab to detect antigen expression, but this is a good idea that we will incorporate into future work.
Minor Comments		

1.	Line 15: specify how was the delivery.	The text has been modified to “We have repurposed an attenuated strain of Salmonella Typhimurium (YS1646) to deliver the receptor binding domains (rbd) of TcdA and TcdB to gut-associated lymphoid tissues (GALT), to elicit a mucosal response against C. difficile. ” (lines 16-19)
2.	Line 16-18: not clear what is meant by cassettes where stably-integrated. Where? how where they regulated in Salmonella ? Please provide more details.	The text has been modified to “In this study, YS1646 candidates with either rbdA or rbdB expression cassettes integrated into the bacterial chromosome at the attTn7 site , were generated ...” (lines 19-20) “The promoters used were either constitutively active (pfrr) or induced when Salmonella is within host cells (ppagC and pSspH2). Two secretory signals were used that drive secretion at different times during bacterial invasion and intracellular residence. SspH2 is specific to the SPI-II T3SS, while SspH1 can be secreted by both SPI-I and SPI-II T3SSs.... Chromosomal integration occurred at the attTn7 site which is located adjacent to the constitutively active glmS gene²⁷. ” (Lines 120-123 and lines 129-130)
3.	Lines 36-37: these should also be in the abstract.	These details are present in the abstract “used in a short-course multimodal vaccination strategy that combined oral delivery (PO) of the YS1646 candidate(s) on days 0, 2 and 4 and intramuscular delivery of recombinant antigen(s) on day 0.” (lines 20-22) The manuscript is unchanged.
4.	Line 37: was oral vaccination a separate treatment than the multimodal? It is difficult to understand as written.	The text has been modified to “Oral vaccination alone completely protected mice from lethal challenge.” (lines 43-44)
5.	Lines 60-64: readers would appreciate more information regarding these vaccines in Phase 2/3 clinical trials. Antigen employed, mode of delivery, company, etc.	These details were provided in the discussion “The field trial of Sanofi’s aluminum-adjuvanted, formalin-inactivated whole toxin vaccine (Cdiffense TM : NTC01887912) was terminated at the interim analysis, when the vaccine efficacy was determined to be -5.2% [95% CI, -104.1 to 43.5] ²⁰ . More recently, Pfizer reported disappointing results for their aluminum hydroxide-adjuvanted, genetically detoxified toxin candidate vaccine (NTC03090191). In this multi-year study of >17,000 subjects, three doses of the vaccine was only 31% effective at preventing primary CDI ²¹ .” (Lines 289-295) No changes were made in the manuscript.
6.	Line 64: what is being meant by high-profile failure? Please rephrase.	The text has been modified to “These failures near the end of development pipeline suggest that novel strategies...” (lines 70-71)
7.	Line 65-68: Include the rationale of neutralizing both toxins here as well.	The text had been modified to “Both toxins contribute to disease and irreversibly glycosylate Rho GTPases in intestinal epithelial cell cytosol leading to the disruption of the cytoskeleton and tight junctions, loss of stress fibers and an overall loss of intestinal barrier function ⁹⁻¹⁴ . The toxins

		are immunogenic and anti-toxin antibodies have strong neutralization activity in vitro ¹⁵ . Currently, the three vaccines that have reached the stage of Phase 2/3 clinical trials have targeted both toxins...” (Lines 63-68)
8.	Lines 89-92: no reference for this statement. Is this part of this work or previous work by the authors?	The appropriate reference is provided at the end of the previous sentence. No change was made.
9.	Lines 100-102: Abbreviation for oral (PO) is confusing; no abbreviation for multimodal vaccination is provided. It is difficult to differentiated between PO and multimodal.	PO is an abbreviation for per os, which means by mouth or oral. For increased clarity, the text has been modified to “Oral vaccination alone (PO) and multimodal vaccination against TcdA completely protected mice from C. difficile challenge.” (lines 109-111)
10.	Line 248: Although authors state that pfr and ppagC are strong promoters, toxin production in the attenuated strain is weak as observed by western blotting.	We agree that the heterologous antigen expression by Western blot is weaker than we expected using these promoters. The text has been modified to “we tried to mitigate the risk by designing inserts with known strong promoters (pfr and ppagC .” Lines 257-258
11.	Fig supplementary 1: The figure legend fails to describe the treatments. What is IM, PO and IM+PO? It is not clear which is the PBS control. Please revise.	The figure legend has been updated with the abbreviations. The PBS control is not shown on the graph, as the data are presented as the fold change of the mean of the experimental group compared to the mean of the PBS group. For clarity, the text has been modified to “All data are shown as the fold change of the mean of secreted cytokines and chemokines in experimental groups compared to the PBS control. ”
12.	Fig supp 2: same comments as in fig S1.	The figure legend has been updated to include the IM abbreviation.
13.	Fig supp 3: same comments as in fig S1.	The figure legend has been updated with the IM abbreviation.
Reviewer #2		
Comments		
1.	Need more information in the introduction on the safety of this YS1646 strain, why and how it is safe.	The manuscript has been changed to address this concern. “Our group is repurposing a strain of S. Typhimurium , YS1646, that was originally designed as a possible cancer therapeutic in the late 1980s. Like its parental strain (YS72), YS1646 has an msbB mutation that leads to production of a modified lipopolysaccharide mutation (LPS) with markedly reduced potential to cause septic shock. After extensive testing in multiple animal models, YS1646 was used in a large phase I clinical trial in subjects with advanced cancer ²⁵ . Although it failed as an anti-cancer ‘drug’, this research demonstrated that YS1646 was safe

		even when injected at doses up to 10^8 intravenously (IV).” (Lines 89-95) Specific mutation details are present in the methods section “Salmonella enterica Typhimurium YS1646 (ΔmsbB2 ΔpurI ΔSuwwan xyl negative; ATCC 202165; ATCC, Manassas, VA)” (Lines 382-383)
2.	Animal survival assays showed combined immunization of ST and RBDs were not better (actually worse) than immunization with RBDs only?	The manuscript has been changed to address this comment. “The positive control used in this study was IM delivery of one or both recombinant antigens (rrbdA and rrbdB) and oral administration of unmanipulated YS1646. This control generated high IgG titers with a sharply skewed IgG1 profile. As expected, based on our previous work and that of others, this vaccine provided good protection against C. difficile challenge^{26, 28-29}. This positive control strategy is similar to several of the C. difficile vaccine candidates that have entered clinical trials. Although repeated doses of these candidates administered IM with adjuvants over several months generated strong serum IgG titers in humans, the two vaccines that advanced to efficacy trials failed to meet their primary endpoint of protection against primary CDI^{16,17,19}. In a small study, Johal et al observed a decrease in IgA⁺ cells in colonic biopsies of patients with CDI that appeared to correlate with disease severity⁴¹. As a result, the apparent ‘superiority’ of protection provided by the IM only vaccination compared to PO only or multimodal vaccination in mice in some of our experiments is unlikely to predict superior efficacy of this traditional approach in the clinic.” (Lines 281-304)
3.	No rationale on avidity assays. What does this mean?	The text has been modified to “To further characterize the IgG elicited by vaccination, we examined the strength of antibody binding (avidity) and subtypes” (Lines 163-164)
4.	No resource information on biotinylated TcdA	The text has been updated “After washing, 1 μg/ml of biotinylated TcdA (biotinylated using ThermoFisher kit, A39257) was incubated for 2 h at room temperature.” (Lines 562-564)
5.	fecal sample (intestinal contents) IgG and IgA should be determined for mucosal vaccination.	Our lab does not currently have an optimised fecal Ig ELISA assay. However, it is an excellent suggestion that we will incorporate in future work. For the current work, we believe that our intestinal IgA ELISA assay (Fig 3C) does address the question of whether our vaccines elicit a mucosal IgA response. The text is unchanged.
6.	IgA and IgM in sera should be determined.	We did not assess serum IgM or IgA for the following reasons. Most IgA produced by mice is specifically targeted to protecting mucosal surfaces, rather than circulating in the serum (Gibbons DL, Mucosal Imm, 2011). Regarding possible IgM responses, we were more interested in the development of durable immunity than the initial response. This is why we sampled the mice 5 weeks after vaccination, giving them time to generate both IgG and IgA responses.

		No changes were made in the manuscript.
7.	More confusing is the mouse model of CDI. Fig4 A and Fig 4B used different challenge dosages without rational? I am also confuse by FigB (challenged with toxins?) Fig C is not useful.	We respectfully disagree with the Reviewer here. The challenge dose of C. difficile used in the experiments presented in Figures 4A and 4B were the same. However, the challenge dose does change between 4ABC and 4DE because the mice sourced from different suppliers were discovered to require different doses to achieve lethality in the control groups. This change in our model was explained in the original text (now Lines 342-348) “It is worth noting that part of our motivation for the changes in the model between our prior studies and the current experiments was an unexpected difference in the apparent sensitivity of the C57BL/6J mice acquired from either Charles River Laboratories (relatively resistant) or Jackson Laboratories (relatively sensitive). We hypothesize that this observation may be attributable to differences between the mice bred in these two facilities in the sensitivity of their gut microbiota to the antibiotic ‘cocktail’ used and possible exposures to non-toxigenic C. difficile strains⁵⁰.” Again, we respectfully disagree with the Reviewer regarding 4C (and 4E). The heatmap, as a visual representation of the health of the mice after challenge, provides additional information regarding how well the mice were protected, that is not well-represented in survival curves. Differences in both morbidity and mortality are both very important outcomes in vaccine studies. The text is unchanged.
8.	sFig 1 is hard to read. Why not use more clearer column figure?	We prefer to present this information as a Figure but can change it to a table if the Journal has a style-guide that favours this format.
9.	sFig 2: one group only used 1 mouse!?	Supplemental Figure 2 shows data in survivors from a challenge experiment in which only one mouse from that group survived. We did not include this group in our analyses. We included it in the graph for full transparency, but if the Journal would prefer, we can remove the group from the graph. We have updated the figure legend to clarify that that group was removed from the analysis. “Data are shown as the mean and SD, with a multiple comparison test comparing all groups, except for the sole surviving mouse vaccinated PO only against both TcdA and TcdB, to the PBS control group (n=1-8, 1 repeat).”
10.	Group size in animal experiment is confusing. 1, 6,.....?	Although the Reviewer’s comment is difficult to understand, we believe this is in reference to Supplemental Figures 2 and 3 in which data from mice that survived challenge are presented. As a result, the final group numbers were out of our control. This is one of the reasons

		we placed these Figures in the Supplemental information, rather than in the manuscript itself. The intrinsic variability of the numbers/group is explained in the Figure legends for both Figures. “Serum of surviving mice was collected 3 weeks after challenge and IgG titers were determined by ELISA.” (S. Fig 2) “The small intestine of surviving mice was collected 3 weeks after challenge and IgA titers were determined by ELISA.” (S. Fig 3) Manuscript is unchanged.
--	--	---

Again, thank you for your consideration of our work.

Sincerely,

Brian J Ward MSc, MDCM, FRCP(C)
Professor of Medicine & Microbiology
Research Institute of the McGill University Health Centre

November 18, 2022

Dr. Brian J Ward
Research Institute of the McGill University Health Centre
1001 Decarie Blvd
Montreal, Quebec
Canada

Re: Spectrum03109-22R1 (Vaccination with an attenuated *Salmonella* Typhimurium vector (YS1646) with chromosomal expression of *Clostridioides difficile* antigens protects mice from lethal challenge)

Dear Dr. Brian J Ward:

Thank you for submitting your manuscript to Microbiology Spectrum.

Your revised manuscript was assessed by an expert reviewer, and as you can see below, they do not feel that their concerns were addressed. I have also reviewed your manuscript and agree with their assessment. I think that additional experimental data needs to be included to improve the technical quality of this work. Unless all points are addressed adequately, I am unable to accept this paper for publication. Please submit a revised manuscript addressing all of the reviewers' concerns.

Link Not Available

Sincerely,

Meera Unnikrishnan

Journals Department
Reviewer comments:

Reviewer #1 (Comments for the Author):

Although the authors have addressed some comments, they have refused to address the major concerns that regard with proper rigor of this work.

Among my major concerns:

Although authors have re-written the title, referring to "C. difficile antigens" is broad and misleading. Correct "antigen" for "receptor binding domains of TcdA and TcdB".

Again, a major concern with this work is the lack of rigor to confirm that chromosomally integrated chimeras and/or the strain itself does not have off mutations. This is not stated in the methods and if strains were indeed whole genome sequence, the illumine reads should be made available in a public domain. Authors need to at least provide evidence of proper integration of the antigens by sequencing technology regardless of proprietary information. There are many tools available that can demonstrate the community that the integration was properly done: WGS of the strain, assembly of contigs, and subsequent demonstration of the alleles containing the antigens; sanger sequencing of the insertion locus; nanopore sequencing of the insertion locus, etc...

Again, another major concern that authors did not address properly regards toxin expression in attenuated Salmonella strains. The western blot shown, barely has detectable immunoreactive bands and lacks proper loading control. For the western blot, a proper loading controls is required.

In addition, authors response to my previous critic that an ELISA proper quantification of chimera production by the different strains is required is not adequate and lacks proper rational. Authors have shown their ability to perform ELISA, and commercial antibodies are available. Given the that the ability of the strains to express the antigen is fundamental to the conclusions of this work, this is an essential control.

Additional concerns that were improperly addressed by the authors

The reference (DOI: <https://doi.org/10.1128/AAC.00360-12>) used for clinical score does not provide a description of how the clinical score was calculated. Please incorporate this in the method section and add appropriate reference.

Provide proper locus tag for the integration site so that non-Salmonella experts can easily identify the site.

Again, authors did not address my previous comment on many references missing. In this context, many statements lack proper references dampening the rigor of the overall manuscript to be publishable. There are too many sections (several sentences stating different facts) that absolutely lack references. This is not acceptable. Below are just a few examples.

Lines 81-87: No reference is provided to none of the sentences.

Lines 89-92: no references is provided

Lines 97-100: no reference.

Again, authors insist in using scientific jargon to define (PO), which is confusing. Instead change for oral administration and its proper abbreviation, term that is widely used in the scientific community.

In my previous critic "Although authors state that pfr and ppagC are strong promoters, toxin production in the attenuated strain is weak as observed by western blotting." Authors again did not properly address this critic; adding "known" to the phrase does not improve the context. An Elisa will conceal this major concern.

Staff Comments:

Preparing Revision Guidelines

Please return the manuscript within 60 days; if you cannot complete the modification within this time period, please contact me. If

you do not wish to modify the manuscript and prefer to submit it to another journal, please notify me of your decision immediately so that the manuscript may be formally withdrawn from consideration by Microbiology Spectrum.

Our responses to the reviewer' comments and suggestions are summarized in the following table with changes in the revised Manuscript indicated where relevant. All simple editorial recommendations have been incorporated into the manuscript; only the more substantive comments are addressed below.

	Comments	Responses
Major Concerns		
1	Although authors have re-written the title, referring to "C. difficile antigens" is broad and misleading. Correct "antigen" for "receptor binding domains of TcdA and TcdB".	The title has been revised to " Multimodal vaccination targeting the receptor binding domains of Clostridioides difficile toxins A and B with an attenuated Salmonella Typhimurium (YS1646) vector protects mice against lethal challenge"
2	Again, a major concern with this work is the lack of rigor to confirm that chromosomally integrated chimeras and/or the strain itself does not have off mutations. This is not stated in the methods and if strains were indeed whole genome sequence, the illumine reads should be made available in a public domain. Authors need to at least provide evidence of proper integration of the antigens by sequencing technology regardless of proprietary information. There are many tools available that can demonstrate the community that the integration was properly done: WGS of the strain, assembly of contigs, and subsequent demonstration of the alleles containing the antigens; sanger sequencing of the insertion locus; nanopore sequencing of the insertion locus, etc..	Whole genome sequencing has been performed on YS1646, YS1646::rbdA, and YS1646::rbdB. We have confirmed the insert follows glmS . The insert and glmS sequence have been included as supplemental files and the whole genome sequence is available upon request. The text has been updated to reflect this. " Successful integration was confirmed by PCR and whole genome sequencing (supplemental files 2-4). " (Lines 135-136)
3	Again, another major concern that authors did not address properly regards toxin expression in attenuated Salmonella	The western blots were performed loading 670 ng of protein from each sample. Figure 1 and the methods have been updated to

	strains. The western blot shown, barely has detectable immunoreactive bands and lacks proper loading control. For the western blot, a proper loading controls is required.	reflect the change. “50µL of Overnight cultures were centrifuged at 21,130xg for 10 min and resuspended in 100µL-PBS. 670 ng of each sample, and 7 ng and 50 ng of positive controls, were then mixed in with NuPAGE lithium dodecyl sulfate (LDS) sample buffer (Invitrogen) according to the manufacturer’s instructions. All samples were heated for 10 min at 100°C and then cooled on ice. 15µl of sample was loaded in each lane; Proteins were separated on a 4 to 20% Bis-Tris protein gel” (Lines 455-460)
4	In addition, authors response to my previous critic that an ELISA proper quantification of chimera production by the different strains is required is not adequate and lacks proper rational. Authors have shown their ability to perform ELISA, and commercial antibodies are available. Given the that the ability of the strains to express the antigen is fundamental to the conclusions of this work, this is an essential control.	We agree with the reviewer that the generally greater sensitivity of the ELISA format might have allowed us to identify additional constructs capable of producing receptor binding domain antigens. However, we disagree that this is ‘an essential control’ since the constructs we carried forward into animal testing produced detectable antigen by western blot. We nonetheless tried to address the Reviewer’s concern using endpoint titer ELISAs to quantify the antigen being produced by the different YS1646 strains. Unfortunately, using commercial polyclonal anti-toxin antibodies, we encountered very high background signal against the YS1646 vector itself, making it impossible to detect antigen production in our strains. Although we could repeat this exercise using monoclonal antibodies directed against the individual toxins, we do not feel this extra effort is justified given detection of antigens by western blot by the strains we carried forward into animal testing.
Additional Concerns		
1	The reference (DOI: https://doi.org/10.1128/AAC.00360-12) used for clinical score does not provide a description of how the clinical score was calculated. Please incorporate	We have updated the reference and included a symptom scoring chart as Supplemental Table 2.

	this in the method section and add appropriate reference.	
2	Provide proper locus tag for the integration site so that non-Salmonella experts can easily identify the site.	The Tn7 integration site does not have a locus tag, so we have provided the locus tag for glmS . "Chromosomal integration occurred at the attTn7 site which is located adjacent to the constitutively active glmS gene (locus_tag: SL1344_3828; FQ312003)" (Lines 130-132)
3	Again, authors did not address my previous comment on many references missing. In this context, many statements lack proper references dampening the rigor of the overall manuscript to be publishable. There are too many sections (several sentences stating different facts) that absolutely lack references. This is not acceptable. Below are just a few examples. Lines 81-87: No reference is provided to none of the sentences. Lines 89-92: no references is provided Lines 97-100: no reference.	We have reviewed the manuscript and added an additional 13 references. When references applied to multiple sentences, we have included the reference at the end of each sentence to increase clarity.
4	Again, authors insist in using scientific jargon to define (PO), which is confusing. Instead change for oral administration and its proper abbreviation, term that is widely used in the scientific community.	We acknowledge that some medical journals have recently begun to use 'oral' rather than the historical ' per os ' (Latin for 'through the mouth' or more simply 'by mouth'). However, we respectfully disagree with the Reviewer that PO is 'scientific jargon' since this shorthand is still widely used in both the medical and scientific literature. Both PO and NPO (ie: 'nil per os' or 'nothing by mouth') continue to be widely used by virtually all English-speaking physicians and in virtually all medical facilities in English-speaking countries. Furthermore, we are not aware of any more 'proper abbreviation' for oral administration alluded to by the Reviewer. We have nonetheless changed all uses of "PO" to "oral" throughout the manuscript.
5	In my previous critic "Although authors state that pfr and ppagC are strong	The text has been changed to highlight the reviewers comments.

	promoters, toxin production in the attenuated strain is weak as observed by western blotting." Authors again did not properly address this critic; adding "known" to the phrase does not improve the context. An Elisa will conceal this major concern.	“Although this strategy likely reduced expression of the targeted antigens, we tried to mitigate the risk by designing inserts with at least one constitutively active promoter known strong promoters (pfrr and ppagC) (36). However, our design still resulted in weak antigen expression. Only two of the strains containing chromosomally-integrated recombinant DNA produced levels of heterologous antigens that were detected by western blot (frr_SspH1_rbdA and pagC_SspH1_rbdB), but both appeared to tolerate the presence of the foreign sequence well, as they had minimal changes in fitness.”
--	--	--

Re: Spectrum03109-22R2 (Multimodal vaccination targeting the receptor binding domains of Clostridioides difficile toxins A and B with an attenuated Salmonella Typhimurium vector (YS1646) protects mice from lethal challenge)

Dear Dr. Brian J Ward:

Your manuscript has been accepted, and I am forwarding it to the ASM production staff for publication. Your paper will first be checked to make sure all elements meet the technical requirements. ASM staff will contact you if anything needs to be revised before copyediting and production can begin. Otherwise, you will be notified when your proofs are ready to be viewed.

Sincerely,
Meera Unnikrishnan
Editor
Microbiology Spectrum